# Radium-223 Treatment of Patients with Metastatic Castration Resistant Prostate Cancer: Biomarkers for Stratification and Response Evaluation

**DOI:** 10.3390/cancers13174346

**Published:** 2021-08-27

**Authors:** Kim van der Zande, Wim J. G. Oyen, Wilbert Zwart, Andries M. Bergman

**Affiliations:** 1Department of Medical Oncology, Netherlands Cancer Institute, Plesmanlaan 121, 1066 CX Amsterdam, The Netherlands; k.vd.zande@nki.nl; 2Division of Oncogenomics, Netherlands Cancer Institute, Plesmanlaan 121, 1066 CX Amsterdam, The Netherlands; 3Department of Nuclear Medicine, Rijnstate Hospital, Wagnerlaan 55, 6815 AD Arnhem, The Netherlands; WOyen@Rijnstate.nl; 4Oncode Institute, 3521 AL Utrecht, The Netherlands

**Keywords:** prostate cancer, metastatic castration resistant prostate cancer, mCRPC, radium-223, alpha-emitting therapy, radiopharmaceuticals, biomarkers, bone metabolism, circulating tumor DNA, inflammation, radiographic evaluation, health-related quality of life

## Abstract

**Simple Summary:**

Radium-223 dichloride ([^223^Ra]RaCl_2_; Ra-223) is an alpha-emitting radiopharmaceutical treatment for patients with metastatic castration resistant prostate cancer (mCRPC) with predominantly bone metastases. While responses to chemotherapeutic and antihormonal mCRPC treatments can be assessed by serum PSA levels, a decrease of serum PSA levels is not expected during Ra-223 therapy. Moreover, radiographic evaluation of bone metastases response is challenging. Therefore, novel biomarkers to select patients for Ra-223 treatment and monitoring response are urgently needed. In this review, we discuss the currently used and exploratory biomarkers for this purpose, including soluble and cellular factors detected in the peripheral blood, genetic defects and radiographic assessments. We conclude that some biomarkers, including metabolic products of collagen degradation and novel PET scan techniques, might hold promise as predictors of response to Ra-223 treatment. However, these biomarkers have not been extensively studied. Consequently, currently, no biomarker has established a place in patient stratification and response evaluation.

**Abstract:**

Radium-223 dichloride ([^223^Ra]RaCl_2_; Ra-223) is a targeted alpha-emitting radiopharmaceutical which results in an overall survival and health related quality of life (HRQoL) benefit in symptomatic patients with metastatic castration resistant prostate cancer (mCRPC) and predominantly bone metastasis. Although effective, options to select patients who will derive treatment benefit and to monitor and predict treatment outcomes are limited. PSA response and radiographic evaluation are commonly used in mCRPC treatment assessment but are not informative in Ra-223 treated patients. Consequently, there is a clear need for predictive and prognostic tools. In this review, we discuss the physiology of bone metastases and the mechanism of action and efficacy of Ra-223 treatment, as well as offering an outline of current innovative prognostic and predictive biomarkers.

## 1. Introduction

Prostate cancer (PCa) is the second most common cancer in men, accounting for 13.5% of all cancer diagnoses in males worldwide [1]. PCa is recognized as an androgen-sensitive disease, since it requires testosterone for its initiation and development [2]. The standard of care for patients diagnosed with metastatic PCa is androgen-deprivation therapy (ADT), which results in the suppression of Androgen Receptor (AR) signaling in PCa cells [3,4]. ADT is administered through suppression of testicular testosterone synthesis by chronic administration of analogues or agonists of gonadotropin-releasing hormone (GnRH) or by bilateral orchiectomy [5,6,7,8]. These interventions result in very low levels of circulating testosterone, to which virtually all patients respond. Inevitably, patients develop progression of the disease after a median of 2–3 years after treatment initiation [7,9,10,11]. This stage of the disease is referred to as metastatic castration-resistant prostate cancer (mCRPC) and represents end-stage disease with significant morbidity and mortality. The hallmark of mCRPC development is the restoration of AR signaling despite continued suppression. Resistance to AR suppression is mediated by various intracellular mechanisms, including AR amplification, mutations, overexpression and splice variants [9].

After an overall survival (OS) benefit was established in mCRPC patients treated with docetaxel in 2004, the number of treatment options for mCRPC patients has been expanding [12]. Currently, multiple agents, i.e., docetaxel, sipuleucel-T, cabazitaxel, abiraterone acetate, enzalutamide and radium-223 dichloride ([^223^Ra]RaCl_2_; Ra-223), have been shown to improve OS of mCRPC patients [13,14,15,16,17,18]. Recently, the Poly (ADP-ribose) polymerase (PARP) inhibitor olaparib and the radioligand Lutetium-177-PSMA-617 were added to this treatment landscape [19,20]. Although effective, resistance to these treatments will invariably develop and, consequently, mCRPC remains a lethal disease. AR amplification and the occurrence of constitutively active AR splice variants are the two best known mechanisms of resistance to the AR Targeted Agents (ARTAs) enzalutamide and abiraterone, while PTEN loss and mutations of p53 and RB are associated with resistance to cabazitaxel and docetaxel [21,22]. As a consequence, cross-resistance between the ARTAs abiraterone, enzalutamide and the newer apalutamide and darolutamide is suggested, but also between chemotherapeutic options, as well as between ARTAs and chemotherapy, which is an important clinical challenge [23,24].

Multiple radio ligand therapies (RLT), including β-emitting samarium-153-EDTMP, strontium-89 dichloride and rhenium-188-HEDP, have shown efficacy as a palliative treatment for patients with symptomatic bone metastases, but an OS benefit has not been established [25]. The alpha-emitting Ra-223 is the only RLT which results in an OS benefit and a delay of skeletal related events (SRE) in mildly symptomatic mCRPC patients with predominantly bone metastases. As a calcimimetic drug targeting osteoblasts, Ra-223 has a unique mechanism of action in the sequence of life-prolonging treatment options [26]. Although clinically effective, significant decreases of serum levels of the biomarker prostate specific antigen (PSA) are rarely established. This is in contrast to patients treated with ARTA or chemotherapy, where PSA is commonly used as a surrogate marker of efficacy. Another challenge is radiological assessment of Ra-223 efficacy, since a response of bone metastases is often difficult or even impossible to establish by conventional bone scan and CT scan. Consequently, challenges in selecting patients for Ra-223 treatment and evaluation of response to Ra-223 treatment remain. In this review, we discuss various aspects of bone metastatic mCRPC and treatment with Ra-223, with a focus on biomarkers for patient stratification and response evaluation.

## 2. Healthy Bone Metabolism

Bone is a dynamic structure consisting of bone cells embedded in bone matrix (collagen and minerals). Bone remodeling is managed by the interaction of bone-resorbing osteoclasts and bone-forming osteoblasts. (Figure 1.) Osteoclasts originate from the myeloid cell lineage, and develop through successive intermediates, including monocytes as a common macrophage and osteoclasts specific progenitors [27]. The differentiation of monocytes into osteoclasts is driven by macrophage colony stimulating factor (M-CSF), receptor activator of nuclear factor kappa-β ligand (RANKL), interleukin-6 (IL-6) and interleukin-8 (IL-8), which are secreted by osteoblasts into the bone micro-environment [28]. Subsequently, osteotropic factors (parathyroid hormone, 1,25-dihydroxyvitamin D3 and prostaglandins) facilitate the maturation process [28,29]. During the bone resorption and collagen degradation process, various metabolic products, including N-telopeptide (NTx), C-telopeptide type-1 collagen (CTx), amino-terminal procollagen propeptides (P1NP) and pyridinoline (PYR), are released [28,30]. Moreover, multiple growth factors are released into the environment including transforming growth factor-beta (TGF-β), bone morphogenetic proteins (BMPs), insulin-like growth factors and platelet derived growth factor (PDGF) [29,31]. These growth factors will activate osteoblast differentiation from stromal mesenchymal stem cells (MSCs), while osteoclasts will undergo apoptosis after bone resorption [32]. Under the influence of BMPs, the canonical Wnt/β-catenin signaling pathway is activated in osteoblasts, and cells will differentiate into osteocytes [32,33]. The remaining osteoblasts that do not mature become inactive or undergo apoptosis [31]. Osteoblasts produce osteocalcin and calcified matrix, while early osteoblast precursors produce the serum biomarker alkaline phosphatase (ALP) [31,32]. Consequently, serum levels of bone-specific ALP (b-ALP) are considered to reflect bone formation. 

The tightly balanced process of bone resorption and formation is managed by the RANK-RANKL-OPG axis [32]. RANKL is produced by osteoblasts and stromal cells, and its expression is controlled by these particular cells and indirectly by osteotropic factors. The Receptor Activator of Nuclear factor Kappa-β (RANK) is located on osteoclast (precursor) cells. RANKL binds to the RANK receptor, which activates osteoclast formation. Consequently, RANK-RANKL activation will result in bone resorption. However, osteoprotegerin (OPG), a decoy receptor for RANK, is produced by osteoblasts and forms a negative feedback loop. The ratio between upregulation of RANKL or OPG determines the level of osteoclast activity [32].

## 3. Bone Metastases in mCRPC

In PCa patients, bone metastases are associated with a shorter life expectancy and bone-related complications including decreased hematopoiesis and SRE [34,35]. SRE is a term that reflects the common complications of bone metastases, including pathologic fractures, spinal cord compression or the need for surgery or radiotherapy to the bone. These complications all have a negative impact on the health-related quality of life (HRQoL) and survival of the patient [34,35,36,37]. Treatment of symptomatic malignant bone disease includes External Beam Radiation Therapy (EBRT), RLT, surgery and analgesics. Bone resorption, the underlying cause of skeletal complications, is prevented by osteoclast targeting bisphosphonates and denosumab, a RANKL directed monoclonal antibody [38,39].

More than 90% of patients with mCPRC will develop metastases to the bone, which is the most frequent site for distant metastases of PCa. [40]. A possible explanation for the preference for bone as a site for metastases was first provided in 1889 by Stephen Paget, and has become known as the seed-and-soil hypothesis [41]. Through the years, this hypothesis has evolved into a better understanding of the interactions of malignant cells (the seeds) and the tumor microenvironment (soil) [42,43]. 

The existence of a premetastatic niche, a supportive environment within a hostile microenvironment, is the basis of this complex interaction. The formation of this premetastatic niche is initiated by the primary tumor that secretes factors into the blood stream that find effectors in specific distant sites [42,43]. As a result, the hostile environment is converted into a safe haven for the circulating tumor cells. Fundamental for this premetastatic niche in bone is the education of osteoblasts and bone marrow cells. This results in relevant changes of the osteoblasts that will boost the chance of successful seeding of PCa cells upon arrival [44,45]. Once located in the bone, metastatic PCa cells will activate the osteoclasts by releasing osteolytic factors (TGF-β, PDGF, VEGF, M-CSF and RANKL (Figure 1)) [44]. In addition, the tumor cells have the ability to mimic normal osteoblast activity (osteomimicry) by releasing the osteoblastic factors osteocalcin, ALP and BMPs (Figure 1) [44]. This disruption of the normal bone metabolism with the release of tumor growth factors results in a positive feedback loop which stimulates the survival and proliferation of PCa cells. This unique environment spins into a vicious cycle of bone degradation and formation [33]. 

In contrast to osteolytic bone metastases, PCa bone metastasis predominantly comprises osteoblastic lesions characterized by the formation of low quality bone tissue. The initial increase in bone formation leads to limited space available for the cancer cells, which confines metastasis development and might explain why osteoblastic lesions progress more slowly than osteolytic lesions [28]. As a reaction to the ongoing osteoblastic activity, osteolysis will emerge and eventually create more room for the growth of metastatic cells [28]. The development of the bone metastasis ultimately results in an impaired and dysregulated bone architecture with loss of strength.

## 4. Radium-223 as a Treatment for Patients with Bone Metastatic mCRPC

Currently, Ra-223 is the only bone-targeted RLT with established OS benefit in patients with mCRPC and bone metastases. Moreover, Ra-223 treatment prevents SREs, and patients experience an improved quality of life benefit from this treatment [26].

### 4.1. Mechanism of Action of Radium-223

Therapeutic bone targeting radiopharmaceuticals can be divided into calcium-analogues such as Ra-223 and Sr-89 and bisphosphonate derivatives such as Sm-153 EDTMP and Re-186-HEDP. Calcium analogues are incorporated into the bone by osteoblasts by the same route as calcium, while bisphosphonate derivatives bind to the osteoid matrix in bone formation [46]. Since metastatic bone lesions induce a high degree of bone turnover with an increase in calcium uptake and construction of bone matrix, radionuclides have a certain specificity for incorporation into metastases-associated bone. Despite this specificity, radiation damage to normal cells will occur, and consequently, common side effects of RLT include damage to healthy bone marrow which is reflected by a decrease in peripheral blood cell counts.

Once incorporated into the bone, Ra-223 decays and produces alpha-particles, which will introduce cytotoxic double-strand breaks (DSBs) to the DNA of the adjacent tumor cells. Alpha particles have a much higher energy transfer and a much shorter range than beta particles, and therefore, have a more favorable bone marrow toxicity profile [26,47,48]. 

### 4.2. Efficacy of Radium-223 in Patients with mCRPC

Bone targeted RLT demonstrated pain relief and an improved quality of life in PCa patients with bone metastases [49]. However, treatment with these drugs does not result in an increase in OS advantage [50]. In the 2013 ALSYMPCA study, mildly symptomatic patients with mCRPC and predominantly bone metastases were randomized between Ra-223 and placebo treatment. Patients treated with Ra-223 showed a significantly improved OS of 3.6 months, which was unique for a radiopharmaceutical therapy [39,51,52]. Moreover, Ra-223 treated patients had significantly fewer SREs and a better quality of life compared to placebo treated patients [26,53]. After publication of the ALSYMPCA trial, Ra-223 was approved by the FDA and EMA as a treatment option for mCRPC patients with symptomatic bone metastases and limited extra osseous metastases [54,55]. However, the results of an early access program suggested a greater benefit in asymptomatic patients than in symptomatic patients in terms of completing more Ra-223 cycles, time to first skeletal event and OS [56]. A prospective observational registry in a nonstudy population of 300 patients evaluated the clinical outcomes of Ra-223 treated patients [37]. In the heterogeneous population of symptomatic and nonsymptomatic patients treated with Ra-223 as various lines of therapy, the median OS was 15.2 months, which was comparable to the OS established in the Ra-223 treated symptomatic patients in the ALSYMPCA trial [37]. Both real-life studies suggest that Ra-223 treatment is effective in symptomatic and asymptomatic patients with mCRPC [37,56].

Although Ra-223 treatment resulted in a better OS, time to first SRE and HRQoL compared to placebo, PSA responses in Ra-223 treated patients were rare and radiological responses were not assessed in ALSYMPCA [26]. Multiple studies evaluated the clinical utility of candidate biomarkers for Ra-223 treatment assessment. Apart from the more commonly used serum biomarkers in PCa care including PSA response, also innovative serum and molecular biomarkers, and radiographic assessments were evaluated for their predictive and prognostic potential. 

Currently, Ra-223 is the only bone-targeted RLT with an established OS benefit in patients with mCRPC and bone metastases. Moreover, Ra-223 treatment prevents SREs, and patients experience an improved quality of life benefit from this treatment [26]. 

## 5. Molecular Markers

DNA damage repair genes safeguard the genomic integrity of cells; homologous recombination (HR) and nonhomologous end joining (NHJ) pathways detect and correct DSBs [57]. While NHJ is error prone, HR is very effective and does not result in a change of genetic information [58]. Inactivating mutations in genes associated with DNA repair mechanisms are frequently found in advanced PCa and are considered drivers of disease progression [57]. The phenotype of cells with inactivating mutations of genes involved in HR is referred to as HR deficiency (HRD). Since the induction of DSBs is the major mode of action of Ra-223 treatment, Ra-223 might be more effective in patients with a HRD, since unrepaired DSBs accumulate faster in HRD cancer cells, which results in cell death [59]. HRD associated gene mutation are found in 10–20% of primary prostate cancers and in 20–30% of mCRPC metastases [60,61,62,63]. With a reported prevalence of approximately 10%, BRCA2 mutations are the most common HRD associated mutations, followed by mutations in BRCA1 and ATM [57,64]. One study suggested that the prevalence of mutations in HR associated genes is similar in the primary tumor and metastatic mCRPC lesions [64]. This suggests that assessments of the primary tumor for mutations associated with HRD are representative of the disease as a whole, and that metastatic biopsies are not needed.

Multiple studies have explored the efficacy of Ra-223 in patients with HRD and mCRPC (Table 1) [65,66,67,68,69]. An early report on four patients with mutations in various HR associated genes demonstrated excellent serum ALP declines, and a case report on a single patient with mutated BRCA2 suggested an exceptionally long response to Ra-223 treatment [66,68]. Data from a small exploratory study in mCRPC patients treated with Ra-223 suggested that patients with HR associated gene mutations had favorable ALP responses and prolonged time to ALP progression in comparison to patients without these mutations, while a more favorable OS could not be confirmed [69]. However, two other studies showed that patients with HRD received more cycles of Ra-223 and had a significantly longer OS than patients without these defects [65,67]. Apart from a possible relationship with response to Ra-223, mutations in HR associated genes are associated with a response to the poly PARP inhibitor olaparib, as was shown in the PROFOUND study [19]. Ongoing trials are investigating the combination of Ra-223 and PARP inhibitors niraparib (ClinicalTrials.gov ID: NCT03076203) and olaparib (ClinicalTrials.gov ID: NCT03317392). 

In conclusion, in patients with an established HRD of the PCa cells, Ra-223 is likely more effective than in patients without these DNA repair defects. Consequently, these genetic hallmarks can be used to select patients for Ra-223 treatment. 

## 6. Serum Biomarkers

To establish the optimal timing of Ra-223 treatment for individual patients and to predict and evaluate responses, both predictive and prognostic markers are essential. Serum markers are commonly used for diagnosis and follow-up of the course of a malignant disease. In PCa patient care, PSA is the most commonly used biomarker. In addition, ALP, a liver enzyme and a marker of osteoblast activity, as well as the more general marker lactate dehydrogenase (LDH), are also applied [70]. Moreover, markers of neuroendocrine differentiation of PCa may be of added value to predict the course of the disease [71]. Novel serum biomarkers have been explored in Ra-223 treated patients with mCRPC, and might hold promise as predictive biomarkers (Table 1). Here, we discuss the commonly used and exploratory serum markers in mCRPC patients treated with Ra-223.

**Table 1 cancers-13-04346-t001:** Overview of suggested biomarkers for patient selection and as prognostic and predictive tools to assess radium-223 therapy.

Biomarker	Mechanism	Patient Selection	Predictive	Prognostic	Ref.
Molecular markers					
Mutations in HR genes	Inability of cancer cells to adequately repair dsDNA breaks	x	x	x	[65,66,67,68,69]
Classic serum biomarkers					
Serum PSA levels	Expressed by (metastatic) prostate cancer cells		x	x	[37,70,72,73]
Serum t-ALP levels	Expressed by osteoblasts precursors and liver cells	x	x	x	[37,70,74]
Serum LDH levels	Predominantly expressed by cancer cells			x	[37,70]
Serum biomarkers of bone-turnover					
Serum b-ALP	Expressed by osteoblasts precursors		x		[75,76]
Serum CTx, NTx, PYR, and P1NP levels	Collagen degradation product		x		[75,76]
Liquid biopsies					
CTC	Prostate cancer cells in peripheral blood		x	x	[59,77,78,79]
γH2AX positive CTC	CTCs with marks of dsDNA damage			x	[59]
Immune biomarkers					
Peripheral blood NLR	Occurrence of immune cells			x	[80,81]
Patient characteristics					
Previous cabazitaxel	Patient characteristic	x	x	x	[37]
Number of Ra-223 cycles	Treatment intensity		x	x	[78,82,83,84,85]
Radiographic imaging					
Bone only disease	Radiographic assessment	x	x	x	[26,37]
Number of bonemetastases	Radiographic assessment		x	x	[26,37]
bone scintigraphyindex	Radiographic estimate of bone metastases burden	x		x	[72,86,87,88]
FDG-PET	Metabolic tumor volume	x		x	[89]
Choline-PET	Tumor volume		x		[90]
18F-NaF PET	Tumor volume		x		[91,92,93]

Patient selection: biomarker possibly useful for selection of patients for Ra-223 therapy, Predictive: biomarker possibly useful for Ra-223 response evaluation, Prognostic marker: biomarker possibly useful to estimate survival. HR: Homologous recombination, PSA: Prostate Specific Antigen, t-ALP: Total alkaline phosphatase, LDH: Lactate dehydrogenase, b-ALP: Bone specific alkaline phosphatase, NTx: N-telopeptide, P1NP: amino-terminal procollagen propeptides, CTC: Circulation tumor cells, NLR: Neutrophil to lymphocyte ratio, Ra-223: Radium-223, FDG-PET: fluorodeoxyglucose- positron emission tomography, 18F-NaF PET: positron emission tomography using 18F-sodium fluoride (NaF).

### 6.1. Classic Prostate Cancer Serum Biomarkers

PSA was introduced in clinics in 1986. Nowadays, it is the most commonly used serum biomarker to assess responses to treatment and for follow-up of the course of mCRPC patients [94]. PSA expression is under the control of androgen receptors, and is produced by the prostate epithelium. Males with healthy prostates only have minimally detectable levels in the bloodstream. Increased levels of PSA are indicative of prostate disorders, which include prostate hypertrophy, infection and malignant disorders [4,6]. PSA reflects the burden of disease in men with mCRPC. Therefore, PSA has a practical utility in informing and updating prognostic information for an individual patient over time [95]. The problem with PSA as a prognostic biomarker lies within the unclear threshold of response, with broad interpatient variance [96]. Moreover, PSA can rise after the start of therapy in a minority of patients; this is referred to as a flare. Therefore, early changes in PSA should not be a reason to stop treatment as advised by the prostate cancer working group (PCWG3), which provides advice regarding assessments of PCa therapies for trial purposes [28]. 

In contrast to other systemic therapies, including docetaxel, abiraterone and enzalutamide, PSA responses are the exception rather than the rule during Ra-223 treatment [73]. Serum PSA unresponsiveness despite an effective treatment might be explained in part by the fact that Ra-223 treatment is mainly directed towards the bone microenvironment rather than to the PCa cells directly. Consequently, the course of serum PSA levels during treatment only provide limited information and should not be considered in Ra-223 therapy decisions [72]. In the ALSYMPCA trial, PSA declines of more than 30% from baseline were observed in 27% of the Ra-223 treated patients [70]. In contrast, in a cohort of 300 real-life patients treated with Ra-223, greater than 30% PSA declines were found in only 6.3% of patients, while this decrease was associated with a better PFS and OS [37]. Conversely, it has been suggested that an increase in serum PSA levels during Ra-223 therapy might be informative for the development of soft-tissue metastases, which are not targeted by Ra-223 [73]. This might suggest that, although rare, PSA responses are of predictive and prognostic value in Ra-223 treated patients with mCRPC.

Total serum alkaline phosphatase (t-ALP) originates from the liver and bone, while b-ALP accounts for 40–50% of serum t-ALP [97]. b-ALP is produced by (precursor) osteoblasts and contributes to the calcification of the bone matrix by mineralization of type 1 collagen [44]. PCa bone metastatic lesions are predominantly osteoblastic, with tumor cells mimicking the normal osteoblast activity [44]. Increased osteoblastic activity indicates an overdrive in bone-formation, or, in the case of PCa, tumor cell growth [74]. t-ALP is often elevated in PCa patients with bone metastases, and is a marker of bone turnover and reflective of PCa bone metastasis growth. It is suggested that the relevance of t-ALP as a prognostic marker is independent of treatment. However, studies suggest that t-ALP performs better as a biomarker of response to bone targeted drugs like Ra-223 [97,98,99]. A subgroup analysis of the ALSYMPCA suggested that patients with a serum t-ALP higher than 220 U/liter derive most benefit from Ra-223 treatment in terms of OS [26]. This suggests that ALP is a prognostic biomarker, which was confirmed in mCRPC patients treated with Ra-223, in whom elevated baseline serum ALP levels were associated with a higher incidence of SREs and shorter than median OS [74]. 

Various studies have described a relationship between changes of serum t-ALP levels during treatment and outcome of Ra-223 treatment [70,100]. When the elevated t-ALP levels decreased <10% from baseline after the first-injection, the OS was significantly shorter compared to patients with a >10% decrease of serum ALP from baseline [70]. Moreover, in one study, a decrease of >10% after first-injection was correlated with a further decrease of >30% from baseline during subsequent Ra-223 cycles [74]. However, other studies did not confirm these findings and described a rapid decrease in t-ALP levels after the first cycles of Ra-223, but an increase after completion of therapy [26,70]. In a real-life cohort, a decrease in serum t-ALP of more than 30% from baseline in mCRPC patients treated with Ra-223 was associated with a longer PFS and OS [37]. These results are in line with the ALSYMPCA trial, where patients with a decrease in serum t-ALP at 12 weeks of treatment had a longer OS than patients that did not have a decrease [70]. In summary, currently available reports available on serum t-ALP levels prior to treatment and changes in serum t-ALP levels during the course of Ra-223 treatment suggest that serum t-ALP acts both as a predictive and prognostic biomarker in patients with mCRPC and bone metastases who have been treated with Ra-223.

LDH is a metabolic enzyme and is involved in the normal glycolysis and gluconeogenesis pathway. It is expressed in every cell of the body, but is particularly active in tumor cells because of aerobic glycolysis pathways that are favored by malignant proliferative cells [95]. Elevated levels of LDH are reflective of underlying tumor burden, but can also reflect tissue necrosis or injury [95]. In the ALSYMPCA trial, elevated levels of LDH accounted for higher risk of death in patients treated with Ra-223, as well as in the placebo treated group [70]. Also in a prospective registry, the level of serum LDH prior to Ra-223 treatment was an independent predictor of OS but not of PFS [37]. Therefore, LDH levels are prognostic for survival but not predictive for Ra-223 treatment outcome. 

### 6.2. Serum Biomarkers of Bone-Turnover

Since the commonly used biomarkers in mCRPC treatment evaluation are of limited value when assessing a response to Ra-223 treatment, alternative biomarkers have been explored that might hold promise as predictive biomarkers. These alternative biomarkers include markers of bone turn over (bone metabolic markers, BMMs). BMMs are by-products of collagen degradation in the bone resorption process and are released into the bloodstream and subsequently into the urine [28]. These markers reflect the rate of formation and resorption of bone in the body as a whole, but not specifically in individual lesions. This might contribute to the fact that, apart from ALP, BMMs have not yet established their utility as predictive biomarkers and are not routinely considered in decision-making [101]. However, it has been suggested that BMMs may be surrogate markers for evaluations of bone-directed therapy [75]. In two studies, the course of serum BMMs levels in Ra-223 treated patients was evaluated [75,76]. 

In a randomized phase II study, patients received four monthly doses of Ra-223 or a placebo [76]. BMMs included t-ALP and b-ALP, P1NP, CTX and CTP, which were measured at baseline and right after treatment completion. After treatment completion, a significant change of serum levels of all five BMMs was found in all Ra-223 treated patients compared to the placebo treated group. Serum b-ALP and P1NP levels showed the biggest reduction, followed by t-ALP and CTx. In contrast, serum CTP levels showed a small increase. The changes of serum BMM levels during Ra-223 treatment were not correlated with clinical outcomes, and therefore, a predictive or prognostic value could not be established. In a more recent randomized trial in patients treated with Ra-223 in combination with enzalutamide or with enzalutamide monotherapy, serum BMM levels were evaluated [75]. In agreement with the previous study, serum levels of P1NP, NTx and b-ALP showed a significant decline after Ra-223 treatment, while CTx levels did not change. Only serum PYR levels also decreased in patients treated with enzalutamide alone. In this study, a change in serum NTx levels was associated with significantly better radiographic response, progression free survival (PFS) and disease control, while changes in serum levels of both P1NP and b-ALP were associated with radiographic PFS [75].

Thus, although serum levels of bone biomarkers do change during Ra-223 therapy, their value as predictive biomarkers has not been unequivocally established. 

### 6.3. Circulating Tumor Cells (CTCs) and Circulating Tumor DNA (ctDNA)

Circulating tumor cells (CTCs) detach from the primary or metastatic tumor and enter the bloodstream, where they can be detected. The concentration of CTCs in the blood is very low in comparison to the background of millions of blood cells. CTC count is considered a prognostic factor in mCRPC patients, as a higher CTC count is associated with a worse outcome [102]. In patients treated with Ra-223, a CTC count ≤5 in 8 mL blood at baseline is predictive for completing the full course of Ra-223 cycles [59,77,78]. Another study suggested that patients with a CTC count <5 have a better OS compared to patients with a CTC count of ≥5 when treated with Ra-223, which underlines its prognostic value [79]. Moreover, the evaluation of CTC counts during Ra-223 suggested that a loss of detectable CTCs at week 9 harbors a potentially favorable prognostic value [59]. In addition to evaluations of CTCs counts, γH2AX positive CTCs have been suggested as a predictive biomarker for risk of death and pain response. When double-strand DNA (dsDNA) breaks occur, γH2AX is produced to recruit proteins involved in DNA repair and chromatin remodeling [59]. Consequently, double-strand break inducing therapy increases the amount of detectable γH2AX in affected cells, and γH2AX positive CTCs might be a biomarker for response to Ra-223 therapy. A small exploratory study in ten patients enumerated γH2AX positive CTCs at baseline and before Ra-223 dose 3 and 6 [59]. A significant increase in γH2AX positive CTCs was related to a lower risk of death. In three patients, a significant increase was seen of γH2AX positive CTCs after a single Ra-223 treatment, which was associated with a significant pain response but no PSA response [59].

Circulating tumor DNA (ctDNA) makes up 1% of all circulating DNA in the blood [103]. ctDNA is an emerging biomarker for solid malignancies; it enables comprehensive tumor genome profiling and is therefore referred to as a ‘liquid biopsy’. It has been shown that all driver mutation in metastatic PCa can be identified in ctDNA, including DNA damage repair defects [103]. Consequently, ctDNA analysis might be applied to identify patients with mCRPC with HRD metastasis that might predict a high vulnerability to Ra-223. However, currently, there are no reports on Ra-223 treated patients selected by ctDNA analysis.

In conclusion, CTC enumeration is predominantly a prognostic biomarker that estimates the disease burden. The selection of patients with a specific mutational landscape, as assessed by ctDNA analysis, is a promising tool; however, clinical data to support this hypothesis is lacking. 

### 6.4. Immune Biomarkers

Introducing DNA double-strand breaks is assumed to be an important element of the cytotoxic actions of Ra-223 treatment. However, the full mechanism of action of Ra-223 is still largely unexplained. Bystander cytotoxicity, in which cell damage occurs by being in close proximity to cells exposed to radiation, is assumed to contribute to the efficacy of Ra-223. Another possible mechanism of Ra-223 activity is the introduction of changes to the so-called tumor microenvironment (TME) as a result of alpha particle radiation [65]. Induced by the occurrence of DNA damage in cells affected by radiation, it has been suggested that TME attracts and activates antigen-specific T-cells (CD8+) which are capable of killing cancer cells, with so called immunogenic cell death as a result. In an exploratory study in 15 patients, CD8+ T cells were isolated from peripheral venous blood before and after Ra-223 therapy [104]. In these cells, the expression of costimulatory and inhibitory molecules CD27, CD28, PD-1 and CTLA-4 were analyzed. Although the overall frequencies of CD8+ T cells did not change during the course of Ra-223 treatment, the frequency of PD-1 expressing CD8+ T cells decreased significantly after Ra-223 treatment [104]. Based on these results, clinical trials are currently exploring combined Ra-223 treatment with the PD-1 targeting immune checkpoint inhibitors pembrolizumab (ClinicalTrials.gov ID: NCT03093428) and atezolizumab (ClinicalTrials.gov ID: NCT03016312). 

Peripheral blood mediators of inflammation hold promise as biomarkers of therapeutic efficacy in various cancers [80]. Of these, neutrophil-to-lymphocyte ratio (NLR) is the most studied cellular inflammation marker, with higher levels being predictive of poor OS [81]. This biomarker has been evaluated in relation to multiple mCRPC treatments, where it showed prognostic value [105]. In one study in 59 patients with mCRPC and treated with Ra-223, a low baseline NLR was independently associated with longer OS [80]. Cytokines and chemokines are important soluble mediators of inflammation, and can be detected in the serum. However, they have not been extensively explored as possible biomarkers of a response to Ra-223 treatment. In a single study, serum levels of the cytokines IFN-y, TNF-alpha and IL-13 were not changed from baseline after completion of Ra-223 treatment [104]. 

Taken together, immune biomarkers have not been extensively explored in patients with mCRPC treated with Ra-223. A single study suggested that NLR might be a prognostic biomarker for response to Ra-223 treatment. 

## 7. Patient Characteristics

The ALSYMPCA trial recruited mCRPC patients with progressive and symptomatic disease with regular use of analgesic medication or recent treatment with EBRT [82]. A subgroup analysis of this trial suggested that there was no difference in OS benefit between patients who used opioid drugs and those who did not. In general, asymptomatic patients have more favorable baseline characteristics in comparison to symptomatic patients, which suggests that these patients have a better prognosis [56]. Indeed, a study in asymptomatic mCRPC patients treated with Ra-223 showed that these patients had better treatment outcomes than symptomatic patients [56]. However, this was not confirmed in a large real-life cohort of Ra-223 treated patients with mCRPC, where being symptomatic or not was not related to PFS or OS [37]. In the latter study, treatment with cabazitaxel prior to Ra-223 was an independent predictor of a worse PFS, which suggests that sequencing of the mCRPC treatment options affect outcome of Ra-223 treatment (Table 1) [37].

The identification of patients who are likely to receive the planned six cycles of Ra-223 can be considered as a both prognostic and predictive biomarker. Multiple studies have shown that patients receiving only one to four cycles of Ra-223 have a shorter OS in comparison to patients receiving five to six cycles [82,83,84,85]. Obviously, the number of Ra-223 cycles received is not an independent biomarker. Patients that received one to four cycles were usually those with a poor baseline status, including a low performance status and baseline hemoglobin [78,84]. Moreover, receiving more Ra-223 cycles was related to better PFS, which suggests predictive biomarker qualities of the number of Ra-223 cycles received. 

To summarize, the selection of patients for Ra-223 therapy who are mildly symptomatic follows the inclusion criteria of ALSYMPCA; however, there is no evidence that these patients benefit more from Ra-223 treatment than asymptomatic patients. Patients not previously treated with cabazitaxel and who are likely to finish more Ra-223 cycles might derive more benefit from treatment. Although biases apply, more Ra-223 cycles are associated with a better PFS and OS. 

## 8. Morphological and Metabolic Imaging

The evaluation of radiological responses by current imaging techniques in patients with predominantly bone metastases is challenging. Consequently, bone metastases are not considered in RECIST response evaluation for clinical trial purposes. Response evaluation by bone scan is complicated by limited specificity and by the “bone flare phenomenon” that may occur early in treatment and should not be confused with progression of disease. This flare is an increase in number of visible lesions despite a clinical response [86,101]. To circumvent this issue, PCWG3 has suggested means to assess progression of bone metastases on a bone scan, but not for response of bone metastases [106].

Patients with at least two bone metastases were included in the ALSYMPCA trial [26]. A subgroup analysis suggested that patients with six or more bone metastases derived an OS benefit from Ra-223 treatment, while those with fewer bone metastases or a super scan did not benefit [26]. Also, in a prospective real-life cohort, the number of bone metastases was found to be an independent risk factor for PFS (Table 1) [37]. Assessment of the tumor burden of the bone, prior and during Ra-223 treatment, might be attractive in predicting and evaluating therapy response (Table 1). Based on a bone scan, a bone scintigraphy index (BSI) is developed to quantify the extent of skeletal tumor burden as the percentage of total skeletal weight. Studies into the value of BSI estimations on interim scans to monitor treatment response have been inconclusive [72,86,87,88]. Consequently, BSI may be less useful for response evaluation, but a low BSI prior to Ra-223 treatment was associated with a favorable OS in two studies, suggesting value as a prognostic marker [72,88].

One study evaluated response to Ra-223 in mCRPC patients by CT and bone scan [107]. During Ra-223 treatment, only 6% developed new bone lesions, while 46% developed extraskeletal progression. In 60% of patients with extraskeletal progression, the sites were not visible on radiological assessments prior to treatment, suggesting that they developed during therapy. These data show that conventional techniques can confirm disease progression but not response to Ra-223 treatment. Since Ra-223 is not expected to have efficacy in extra-osseous metastatic sites, it is advisable to evaluate the development of these lesions during Ra-223 therapy. Patients who have extraskeletal disease should not be considered for Ra-223 treatment, which is in line with the ALSYMPCA trial. In the ALSYMPCA trial, patients with more than a single malignant lymphadenopathy greater than 3 cm and visceral metastases were excluded from trial participation. 

The limited accuracy of bone scintigraphy and CT-scans in the evaluation of bone lesions response on Ra-223 treatment warrants the exploration of alternative radiographic assessments. Novel advanced tools for the evaluation of bone metastases should provide earlier signaling of treatment response or resistance, which would enable earlier change of treatment course and prevention of unnecessary toxicities [101]. Novel tools include the various Positron Emission Tomography (PET) techniques. [^18^]F-Fluorodeoxyglucose (FDG)-PET scans of 28 mCRPC patients treated with Ra-223 were retrospectively analyzed for prognostic stratification. The metabolic tumor volume prior to treatment, as assessed by FDG-PET, was found to be an independent predictor of OS, while patients with a decrease in metabolic tumor volume had a better OS [89]. A recent study assessed the predictive value of [^18^]F-Fluorocholine (FCH) PET/CT imaging combined with bone scans at baseline and during Ra-223 treatment [91]. Although agreement between the two imaging methods at interim and end-of-treatment was weak, the FCH PET/CT scan assessed responses correlated with PFS, while FCH PET/CT and bone scan assessed responses were associated with OS [91]. Based on this study, the [^18^]F-Fluorocholine PET/CT imaging could possibly be of predictive value during and after Ra-223 treatment. Prostate Specific Membrane Antigen (PSMA) targeted imaging has emerged as a promising molecular imaging tool. Responses to various mCRPC therapies were assessed in a retrospective study where [^68^]Ga-PSMA PET scans were made prior to and after completion of therapy [90]. In the nine patients treated with Ra-223, no relationship between [^68^]Ga-PSMA PET response and PSA response was found, which might be related to the low number of Ra-223 treated patients in the study [90]. More extensive research has been undertaken into [^18^]F-NaF PET/CT as a tool to select patients for Ra-223 treatment and to assess response to treatment. In a study with six patients with mCRPC, a [^18^]F-NaF PET/CT was made prior to Ra-223 treatment and after the third cycle, and changes in the uptake values of the tracer in individual bone metastases were evaluated [93]. The intensity of staining prior to treatment was an independent predictor of change of tracer uptake, but no correlations were made with PFS or OS. In a small study, all 10 patients had a [^18^]F-NaF PET response after six Ra-223 cycles, which was related to serum PSA response, suggesting predictive significance [92]. 

To summarize, the number of bone metastases as assessed by bone scan might be useful to select patients for Ra-223 treatment. Although there is agreement on how the progression of bone metastases can be reliably assessed by bone scan, no such consensus exists regarding response to Ra-223 treatment. Several PET based techniques might be of value for response evaluation [91].

## 9. Conclusions and Future Perspectives

Ra-223 treatment is a valuable addition to the life-prolonging treatment options for patients with mCRPC. As a bone targeting agent, it holds a unique place next to ARTAs and chemotherapeutic drugs, which enables a rationale-based sequencing of treatment options. However, Ra-223 treatment is indicated for selected patients. The ALSYMPCA trial included mildly symptomatic patients with predominantly bone metastases in whom it established a benefit in OS and the occurrence of SREs [82]. Considering its mode of action, Ra-223 monotherapy is not expected to be an effective drug in patients with lymph node or visceral metastases. Although the efficacy of Ra-223 was established in a randomized, placebo-controlled trial in patients with mCRPC, response to this drug in an individual patient is difficult to evaluate. PSA responses are rare in Ra-223 treated patients, and classic radiographic evaluation by bone and CT scan can establish a progression of bone metastatic disease, but no response. Therefore, alternative means to select patients for Ra-223 treatment and assessments of response are needed [37,70,72]. In this review, we summarized the potential biomarkers and clinical variables which might hold promise as tools for patient selection and evaluation of Ra-223 response in patients with mCRPC. Biomarkers that hold predictive value based on the course of the biomarker during treatment can be used to support decisions to continue treatment. However, most biomarkers are prognostic and predict the course of the disease independent of treatment. 

Several biomarkers that can be assessed prior to treatment have been suggested to have predictive potential. These biomarkers can be used for patient stratification, and include HRD of PCa cells and levels of serum t-ALP. Moreover, several patient characteristics hold predictive potential, including bone only disease and absence of previous treatment with cabazitaxel. Other predictive biomarkers represent a change in treatment and are mostly metabolic products of collagen degradation [75,76]. However, the limited number of studies on their predictive value in Ra-223 treated patients showed contradictory results. Moreover, change in intensity of bone metastases as assessed by various PET techniques might hold promise as a predictive biomarker. Other biomarkers hold prognostic potential, including serum LDH levels, CTC counts, bone scintigraphy index and FDG-PET/CT assessment. Most likely, these biomarkers reflect the disease burden. Also, the NLR might reflect a more general response to the metastatic disease but does not provide information on the vulnerability of bone metastases to alpha radiation. 

Although potentially of value, none of the above-mentioned biomarkers has established a definitive place in decision making for Ra-223 treated patients. Claims regarding their predictive and prognostic value are mostly based on a limited number of small studies. Moreover, some reports are not unequivocal. Given the paucity of knowledge of the exact mechanism of action of Ra-223, new potential biomarkers might emerge when we learn more about its mode of action. For instance, the reaction of the tumor microenvironment of the bone metastases to Ra-223 treatment and subsequent immune response have not been extensively explored. Assessments of serum levels of mediators of inflammation and infection, such as chemokines and cytokines, in relation to Ra-223 treatment, could be an interesting means by which to understand the mechanism of action of Ra-223, and could be explored as a biomarker [80,104]. 

However, the selection of patients with mCRPC for Ra-223 treatment based on biomarkers might imply the danger of overlooking patients that could benefit from Ra-223 treatment. No biomarker is perfect; they all have their limitations in the sense of specificity and sensitivity. Therefore, optimal selection for Ra-223 treatment and response assessment might be achieved by the use of combinations of multiple biomarkers.

In conclusion, although Ra-223 is a treatment for mCRPC patients associated with improved OS and quality of life, patient selection for this treatment and response evaluation remain challenging. Various biomarkers, including molecular aberrations, serum levels of collagen metabolism products and soluble factors secreted by osteoblasts, as well as patient characteristics and radiographic assessments, have been suggested to support decision making. However, none of these biomarkers has been extensively studied, and consequently, none has established a place in patient stratification and response evaluation. Larger trials into the predictive value of promising and novel biomarkers are needed to optimize Ra-223 treatment.

## Figures and Tables

**Figure 1 cancers-13-04346-f001:**
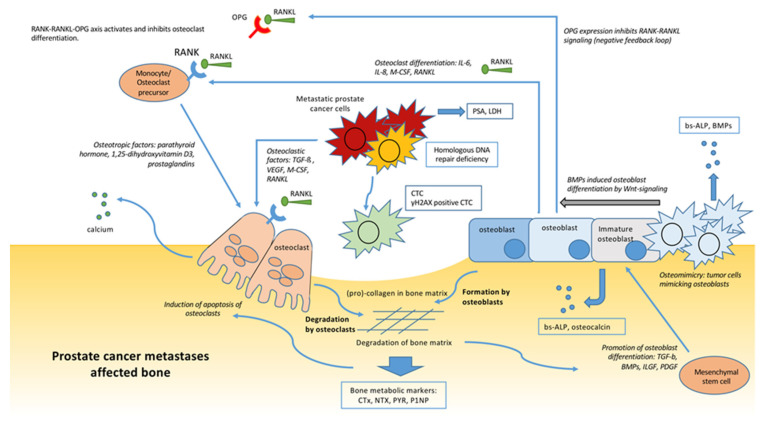
Origin of suggested molecular and soluble markers for selection and assessment of Ra-223 treated patients. Metastatic prostate cancer cells in the bone enhances the vicious cycle of bone degradation and formation. Boxed factors might serve as a biomarker for Ra-223 treatment assessment. Ra-223: radium-223 dichloride, RANK: receptor activator of nuclear factor kappa-β, RANKL: RANK Ligand, OPG: osteoprotegerin, IL-6: Interleukin 6, IL-8: Interleukin 8, M-CSF: Macrophage-colony stimulating factor, PSA: Prostate specific antigen, LDH: Lactate dehydrogenase, TGF-β: Transforming growth factor-beta, VEGF: Vascular endothelial growth factor, CTC: Circulating tumor cell, CTx: C-telopeptide type-1 collagen, NTx: N-telopeptide, PYR: pyridinoline, P1NP: amino-terminal procollagen propeptides, b-ALP: Bone specific alkaline phosphatase, BMP: bone morphogenetic proteins, ILGF: Insulin-like growth factors, PDGF: Platelet-derived growth factors.

## Data Availability

No new data were created or analyzed in this study. Data sharing is not applicable to this article.

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
