# Peer review of "Radium-223 Treatment of Patients with Metastatic Castration Resistant Prostate Cancer: Biomarkers for Stratification and Response Evaluation"

_cancers, 2021, doi:10.3390/cancers13174346_

Round 1

Reviewer 1 Report

Well organized, clear and precise manuscript.
Radium-223 dichloride is the unique bone-directed treatment option in mCRPC. No biomarker, to date, is able to predict the optimal use of Rad- 223 treatment.  There is an urgent need for novel biomarkers ( predictive and prognostic) to select patients for Rad 223.

In this review the authors describe with accuracy the physiology of bone metastases and make an overview of innovative prognostic and predictive biomarkers. I consider the topic relevant in the field, because we have a treatment that improve OS in patients with mCRPC, but, very often, we don't use this arm in real word or we select the patients in the wrong way. No specific improvements regarding methodology ( this is not a systematic review).

The conclusions are consistent with the evidence and arguments presented . It is very important to underline that RSìad -223 is indicated for selected patients (without visceral and lymphnodes metastases). Probably Pet-based techiniques could be useful in in response evaluation, but also in a correct selection of patients.

Furthermore, the references are also appropriate.

Author Response

We thank the reviewer for evaluating our work, and we are delighted to hear that the reviewer appreciates our review on biomarkers for stratification and response evaluation of Radium-223 dichloride ([223Ra]RaCl2; Ra-223) treated patients with mCRPC. The reviewer is absolutely correct to state that patient selection for this treatment is key, yet challenging. Therefore, there is a unmet need to develop both prognostic and predictive biomarkers for Ra-223 treatment.

We agree that Ra-223 treatment is an effective therapy in a selected patient population with predominantly ‘bone metastases’, so without visceral and no or limited lymph node metastases involvement, as is lined out at several locations in the manuscript.

Furthermore, we fully agree that PET-based techniques are promising tools to select patients for Ra-223 treatment and to evaluate therapy response as stated in paragraph 8.

Again, we would like to thank the reviewer for the constructive feedback, and hope the reviewer considers the revised version sufficiently improved to render it acceptable for publication. 

Reviewer 2 Report

This is a complete, up-to-date and very well written review. 

The authors extensively analyzed the current literature on the topic and critically discussed the future prospects of Ra-223 RLT in mCRPC.

No major or minor revisions are required from my point of view.

Just a few minor typos to be checked before accepting it:

  • correct nomenclature should be used throughtout the manuscript for radium, i.e., [223Ra]RaCl2
  • page 11, line 482 "radium-22" instead of "radium-223"
  • page 13, line 587 "overall survival" should be abbreviated with OS, as in the previous paragraphs 

Author Response

We thank the reviewer for evaluating our work, and we are delighted to hear that the reviewer appreciates our review on biomarkers for stratification and response evaluation of Radium-223 dichloride ([223Ra]RaCl2; Ra-223) treated patients with mCRPC. We fully agree with the reviewer that the correct nomenclature for radium-223 is [223Ra]RaCl2. At the first naming of [223Ra]RaCl2 in the ‘Simple summary’, the ‘Abstract’ and the ‘Introduction’, we present the drug now as ‘Radium-223 dichloride ([223Ra]RaCl2; Ra-223)’. Further on in the manuscript, [223Ra]RaCl2 is abbreviated as ‘Ra-223’, which improves readability and is a commonly used abbreviation in literature.

We excuse for the typos and thank the reviewer for pinpointing them out. All have now been corrected (page 11; line 499 and page 13, line 605).

Again, we would like to thank the reviewer for the constructive feedback, and hope the reviewer considers the revised version sufficiently improved to render it acceptable for publication. 

Reviewer 3 Report

Authors present a well-written and informative review on biomarkers for stratification and response evaluation. Recently few reviews focus on the evaluation of these predictive and prognostic markers around PRRT, Ra223-therapy or PSMA-therapy. This is an important issue as especially for 223Ra therapy standard parameter as PSA are of limited value compared to other treatments in mCRPC patients.

 I appreciate the in detail description of healthy bone metabolism and of the mechanism of action of Ra223.

However, I have some minor points to improve:

  • Some phrases are relatively long and/or complicated. Please review the manuscript and rephrase, for example : Page 8 ll 289-292 or ll 295-297.

  • Introduce abbreviation when first mentioned in the text, e.g. PARP (already in the introduction)

  • Heading 3. Should probably not be the same as heading 2. Please change.

  • Page 5 ll189-191: the sentence is misleading as also 177Lu PSMA is a beta emitting radiopharmaceutical. Please specifiy as you have done before (“bone targeted RLT” ll 170.).

  • Heading 8. Should be shorter, as all the other headings. E.g. Morphological and metabolic imaging. This section is not only on response prediction but also on the potential prognostic value.

Style of table 1: Patient se-lection: selection should be in the line below; description of mechanisms difficult to read as to close to the next description and many word separations

Author Response

We thank the reviewer for evaluating our work, and we are delighted to hear that the reviewer appreciates our review on biomarkers for stratification and response evaluation of Radium-223 dichloride ([223Ra]RaCl2; Ra-223) treated patients with mCRPC. However, the reviewer raises several minor points to improve the manuscript. We have incorporated the reviewer‘s suggestions as follows:

#1 Some phrases are relatively long and/or complicated. Please review the manuscript and rephrase. For example: Page * ll 289-292 or II 295-297.

Response: We thank the reviewer for this comment, which will improve the readability of the review. Apart from the long sentences indicated by the reviewer, we traced several more longer sentences which were reviewed and shortened (lines 293-294, 299-300, 309-310, 331, 336-367, 518).

#2 Introduce abbreviations when first mentioned in the text, e.g. PARP (already in the introduction)

Response: We now made sure that abbreviations were introduced when first mentioned in the text (PARP in the introduction and RANK-RANKL, interleukin-6/8 in paragraph 2). Moreover, ‘PCa’ as an abbreviation of ‘prostate cancer’ is now consistently used throughout the manuscript.

#3 Heading 3 should probably not be the same as heading 2. Please change.

Response:  We thank the reviewer for detecting this flaw. We rectified the heading of paragraph 3 to the correct “bone metastases in mCRPC”.

#4 Page 5 ll 189-191: the sentence is misleading as also 177Lu PSMA us a beta emitting radiopharmaceutical. Please specify as you have done before (‘bone targeted RLT’ ll 170)

Response: We changed ‘beta-emitting radiopharmaceuticals’ into ‘bone targeted RLT’ in this sentence, as suggested by the reviewer (line 198).

#5 Heading 8. Should be shorter, as all the other headings. E.g. Morphological and metabolic imaging. This section is not only on response prediction but also on the potential prognostic value.

Response: We thank the reviewer for this suggestion. We shortened the heading of paragraph 8 to an all-encompassing ”morphological and metabolic imaging”” (page 11, line 491), as suggested by the reviewer.

#6 Style of table 1: Patient selection: selection should be in the line below; description of mechanisms is difficult to read as to close to the next description and many word separations.

Response: The lay-out and formatting of table 1 was revised to improve readability. ‘Selection’ in ‘Patient selection’ (third column from left) is now in the line below. Furthermore, the ‘Mechanism’ column (second from left), is now wider and the font size has been changed. Consequently, the short description of the mechanisms of the novel biomarkers are better to read. 

Again, we would like to thank the reviewer for the constructive feedback, and hope the reviewer considers the revised version sufficiently improved to render it acceptable for publication.